# Energy consumption and cooperation for optimal sensing

Vudtiwat Ngampruetikorn[1,2 ✉], David J. Schwab[2,3,6] & Greg J. Stephens[4,5,6]

The reliable detection of environmental molecules in the presence of noise is an important cellular function, yet the underlying computational mechanisms are not well understood. We introduce a model of two interacting sensors which allows for the principled exploration of signal statistics, cooperation strategies and the role of energy consumption in optimal sensing, quantified through the mutual information between the signal and the sensors. Here we report that in general the optimal sensing strategy depends both on the noise level and the statistics of the signals. For joint, correlated signals, energy consuming (nonequilibrium), asymmetric couplings result in maximum information gain in the low-noise, high-signal-correlation limit. Surprisingly we also find that energy consumption is not always required for optimal sensing. We generalise our model to incorporate time integration of the sensor state by a population of readout molecules, and demonstrate that sensor interaction and energy consumption remain important for optimal sensing.

---

[1] Department of Physics and Astronomy, Northwestern University, Evanston, IL 60208, USA. [2] Initiative for the Theoretical Sciences, The Graduate Center, CUNY, New York, NY 10016, USA. [3] Center for the Physics of Biological Function, Princeton/CUNY, New York, USA. [4] Biological Physics Theory Unit, OIST Graduate University, Okinawa 904-0495, Japan. [5] Department of Physics and Astronomy, Vrije Universiteit, 1081HV Amsterdam, The Netherlands. [6]These authors contributed equally: David J. Schwab, Greg J. Stephens. ✉email: vngampruetikorn@gc.cuny.edu

Cells are surrounded by a cocktail of chemicals, which carry important information, such as the number of nearby cells, the presence of foreign material, and the location of food sources and toxin. The ability to reliably measure chemical concentrations is thus essential to cellular function. In fact, cells can exhibit extremely high sensitivity in chemical sensing, for example, our immune response can be triggered by only one foreign ligand[1] and *Escherichia coli* chemotaxis responds to nanomolar changes in chemical concentration[2]. But how does cellular machinery achieve such sensitivity?

One strategy is to consume energy: molecular motors metabolise ATPs to drive cell movement and cell division, and kinetic proofreading employs nonequilibrium biochemical networks to increase enzyme–substrate specificity[3]. Indeed, the role of energy consumption in enhancing the sensitivity of chemosensing is the subject of several studies[4–8]. However, whether nonequilibrium sensing can supersede equilibrium limits to performance is unknown[9,10].

Interactions also directly influence sensitivity, and receptor cooperativity is a biologically plausible strategy for suppressing noise[11–13]. These results, however, apply in steady state[11] and it is independent receptor that maximise the signal-to-noise ratio under a finite integration time[14,15] even when receptor interactions are coupled to energy consumption[16]. More generally, a trade-off exists between noise-reduction and available resources, such as integration time and the number of readout molecules[6,7]. It is therefore important to examine how sensor circuit sensitivity depends on the level of noise and the structure of the signals without a priori fixing the interactions or the energy consumption.

We introduce a general model for nonequilibrium coupled binary sensors. Specialising to the case of two sensors, we obtain the steady state distribution of the two-sensor states for a specified signal. We then determine the sensing strategy that maximises the mutual information for a given noise level and signal prior. We find that the optimal sensing strategy depends on both the noise level and signal statistics. In particular, energy consumption can improve sensing performance in the low-noise, high-signal-correlation limit but is not always required for optimal sensing. Finally, we generalise our model to include time averaging of the sensor state by a population of readout molecules, and show that optimal sensing remains reliant on sensor interaction and energy consumption.

## Results

**Model overview.** We consider a simple system of two information-processing units (sensors), an abstraction of a pair of coupled chemoreceptors or two transcriptional regulations with cross-feedback (Fig. 1a). The sensor states depend on noises, signals (e.g., chemical changes) and sensor interactions, which can couple to energy consumption. Instead of the signal-to-noise ratio, we use the mutual information between the signals and the states of the system as the measure of sensing performance. Physically, the mutual information corresponds to the reduction in the uncertainty (entropy) of the signal (input) once the system state (output) is known. In the absence of signal integration, the mutual information between the signals and sensors is also the maximum information the system can learn about the signals as noisy downstream networks can only further degrade the signals. However, computing mutual information requires the knowledge of the prior distribution of the signals. Importantly, the prior encodes some of the information about the signal, e.g., signals could be more likely to take certain values or drawn from a set of discrete values. Although the signal prior in cellular sensing is generally unknown, one simple, physically plausible

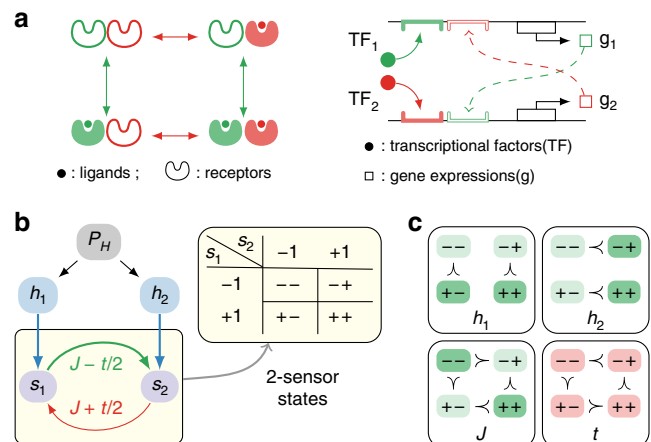

**Fig. 1 Model overview.** We consider a minimal model of a sensory complex, which includes both external signals and interactions between the sensors. **a** Physical examples of a sensor system with two interacting sensors: a pair of chemoreceptors with receptor coupling (left) and a pair of transcriptional regulations of gene expressions with cross-feedback (right). **b** Model schematic. Each sensor is endowed with binary states $s_1 = \pm 1$ and $s_2 = \pm 1$, so that the sensor complex admits four states: $--$, $-+$, $+-$ and $++$. A signal $H$ is drawn from the prior distribution $P_H$ and couples to each sensor via the local fields $h_1$ and $h_2$. The coupling between the sensors is described by $J_{12} = J + t/2$ and $J_{21} = J - t/2$, and can be asymmetric so in general $J_{12} \neq J_{21}$. **c** The field $h_i$ favours the states with $s_i = +1$ (top row); the coupling $J$ favours the correlated states $--$ and $++$ (bottom left); and the nonequilibrium drive $t$ generates a cyclic bias (bottom right).

choice is the Gaussian distribution, which is the least informative distribution for a given mean and variance.

**Nonequilibrium coupled sensors.** We provide an overview of our model in Fig. 1b. Here, a sensor complex is a network of interacting sensors, each endowed with binary states $s = \pm 1$, e.g., whether a receptor or gene regulation is active. The state of each sensor depends on that of the others through interactions, and on the local bias fields generated by a signal; for example, an increase in ligand concentration favours the occupied state of a chemoreceptor. Owing to noise, the sensor states are not deterministic so that the probability of every state is finite. We encode the effects of signals, interactions and intrinsic noise in the inversion rate—the rate at which a sensor switches its state. We define the inversion rate for the $i$th sensor

$$\Gamma^i_{S|H} \equiv \mathcal{N}_H \exp\left[-\beta\left(h_i s_i + \sum_j^{j \neq i} J_{ij} s_i s_j\right)\right], \quad (1)$$

where $S = \{s_i\}$ denotes the present state of the sensor system, $H = \{h_i\}$ the signal, $J_{ij}$ the interactions, and $\beta$ the sensor reliability (i.e., the inverse intrinsic noise level). The transition rate determines the lifetime, and thus the likelihood, of each state $S$. In the above form, the coupling to the signal, $h_i s_i$, favours alignment between the sensor $s_i$ and the signal $h_i$, whereas the interaction $J_{ij} > 0$ ($J_{ij} < 0$) encourages correlation (anticorrelation) between the sensors $s_i$ and $s_j$. The constant $\mathcal{N}_H$ sets the overall timescale but drops out in steady state, which is characterised by the ratios of the transition rates. In the context of chemosensing, the signal $\{h_i\}$ parametrises the concentration change of one type of ligand when all sensors in the sensing complex respond to the same chemical, and of multiple ligands when the sensors exhibit different ligand specificity.

Given an input signal $H$, the conditional probability of the states of the sensor complex in steady state $P_{S|H}$ is obtained by balancing the probability flows into and out of each state while conserving the total probability $\sum_S P_{S|H} = 1$,

$$\sum_i \left[ P_{S^i|H} \Gamma^i_{S^i|H} - P_{S|H} \Gamma^i_{S|H} \right] = 0. \qquad (2)$$

Here and in the following, the state vector $S^i$ is related to $S$ by the inversion of the sensor $i$, $s_i \to -s_i$, whereas all other sensors remain in the same configuration.

In equilibrium, detailed balance imposes an additional constraint forbidding net probability flow between any two states,

$$P^{eq}_{S^i|H} \Gamma^{i,eq}_{S^i|H} - P^{eq}_{S|H} \Gamma^{i,eq}_{S|H} = 0, \qquad (3)$$

and this condition can only be satisfied by symmetric interactions $J_{ij} = J_{ji}$ (see, Coupling symmetry and detailed balance in Methods). We define the equilibrium free energy

$$F_{S|H} = -\sum_i h_i s_i - \sum_{i,j}^{i<j} J_{ij} s_i s_j, \qquad (4)$$

such that the inversion rate depends on the initial and final states of the system only through the change in free energy

$$\Gamma^{i,eq}_{S|H} = \mathcal{N}_H \exp\left[ -\frac{1}{2}\beta\left( F_{S^i|H} - F_{S|H} \right) \right]. \qquad (5)$$

Together with the detailed balance condition (Eq. (3)), this equation leads directly to the Boltzmann distribution $P^{eq}_{S|H} = e^{-\beta F_{S|H}} / \mathcal{Z}^{eq}_H$ with the partition function $\mathcal{Z}^{eq}_H$. When constrained to equilibrium couplings, this model has been previously investigated in the context of optimal coding by a network of spiking neurons[17]. Asymmetric interactions $J_{ij} \neq J_{ji}$ break detailed balance, resulting in a nonequilibrium steady state (see, Coupling symmetry and detailed balance in Methods).

We specialise to the case of two coupled sensors $S = (s_1, s_2)$, belonging to one of the four states: $--$, $-+$, $++$ and $+-$ (Fig. 1b). For convenience, we introduce two new variables, the coupling $J$ and nonequilibrium drive $t$, and parametrise $J_{12}$ and $J_{21}$ such that $J_{21} = J - t/2$ and $J_{12} = J + t/2$ (Fig. 1b). The effects of the bias fields $(h_1, h_2)$, coupling $J$ and nonequilibrium drive $t$ are summarised in Fig. 1c. Compared with the equilibrium inversion rate [Eq. (5)], a finite nonequilibrium drive leads to a modification of the form

$$\Gamma^i_{S|H} = \begin{cases} e^{\frac{1}{2}\beta t} \Gamma^{i,eq}_{S|H} & \text{for cyclic } S \to S^i, \\ e^{-\frac{1}{2}\beta t} \Gamma^{i,eq}_{S|H} & \text{for anticyclic } S \to S^i, \end{cases} \qquad (6)$$

where $S \to S^i$ is cyclic if it corresponds to one of the transitions in the cycle $-- \to -+ \to ++ \to +- \to --$, and anticyclic otherwise. Recalling that this probability flow vanishes in equilibrium, it is easy to see that, depending on whether $t$ is positive or negative, the nonequilibrium inversion rates result in either cyclic or anticyclic steady state probability flow.

A net probability flow in steady state leads to power dissipation. By analogy with Eq. (5), we write down the effective change in free energy of a transition $S \to S^i$,

$$\Delta F^{eff}_{S \to S^i} = \Delta F^{eq}_{S \to S^i} - \begin{cases} t & \text{for cyclic } S \to S^i, \\ -t & \text{for anticyclic } S \to S^i. \end{cases}$$

That is, the system loses energy of $4t$ per complete cycle. To conserve total energy, the sensor complex must consume the same amount of energy it dissipates to the environment. The nonequilibrium drive also modifies the steady state probability distribution. Solving Eq. (2), we have (see also, Steady state

master equation in Methods)

$$P_{S|H} = \exp\left[ -\beta\left( F_{S|H} + \delta F_{S|H} \right) \right] / \mathcal{Z}_H, \qquad (7)$$

where $F_{S|H}$ denotes the free energy in equilibrium [Eq. (4)]. The nonequilibrium effects are encoded in the noise-dependent term

$$\delta F_{S|H} = -\frac{1}{\beta} \ln\left[ e^{\frac{1}{2}\beta t s_1 s_2} \frac{\cosh[\beta(h_1 - t s_2)]}{\cosh\beta h_1 + \cosh\beta h_2} \right. \\ \left. + e^{-\frac{1}{2}\beta t s_1 s_2} \frac{\cosh[\beta(h_2 + t s_1)]}{\cosh\beta h_1 + \cosh\beta h_2} \right], \qquad (8)$$

and note that $\delta F_{S|H} \to 0$ as $t \to 0$.

**Mutual information**. We quantify sensing performance through the mutual information between the signal and sensor complex $I(S; H)$, which measures the reduction in the uncertainty (entropy) in the signal $H$ once the system state $S$ is known and vice versa. For convenience, we introduce the "output" and "noise" entropies where output entropy is the entropy of the two-sensor state distribution $\mathcal{S}[P_S] = \mathcal{S}[\sum_H P_H P_{S|H}]$, whereas the noise entropy is defined as the average entropy of the conditional probability of sensor states $\sum_H P_H \mathcal{S}[P_{S|H}]$. Here, $P_H$ is the prior distribution from which a signal is drawn and the entropy of a distribution is defined by $\mathcal{S}[P_X] = -\sum_X P_X \log_2 P_X$. In terms of the output and noise entropies, the mutual information is given by

$$I(S; H) = \underbrace{\mathcal{S}\left[\sum_H P_H P_{S|H}\right]}_{\text{Output entropy}} - \underbrace{\sum_H P_H \mathcal{S}\left[P_{S|H}\right]}_{\text{Noise entropy}}, \qquad (9)$$

and we seek the sensing strategy (the coupling $J$ and nonequilibrium drive $t$) that maximises the mutual information for given reliability $\beta$ and signal priors $P_H$. In practice, we solve this optimisation problem by a numerical search in the $J$–$t$ parameter space using standard numerical-analysis software (see, Code availability for an example code for numerical optimisation of mutual information).

**Correlated signals**. The bias fields at two sensors are generally different, for example, chemoreceptors with distinct ligand specificity or exposure, and we consider signals $H = (h_1, h_2)$, drawn from a correlated bivariate Gaussian distribution (Fig. 2a),

$$P_H = \frac{1}{2\pi\sqrt{1 - \alpha^2}} \exp\left( -\frac{h_1^2 - 2\alpha h_1 h_2 + h_2^2}{2(1 - \alpha^2)} \right), \qquad (10)$$

where $\alpha \in [-1, 1]$ is the correlation between $h_1$ and $h_2$. When we maximise the mutual information in the $J$–$t$ parameter space, we find that the mutual information is maximised by an equilibrium system ($t^* = 0$) for small $\beta$. In Fig. 2b, we show that the optimal strategy is cooperative ($J > 0$) at small $\beta$ and switches to antic-ooperative ($J < 0$) around $\beta \sim 1$. Below a certain value of $\beta$, the optimal coupling diverges $J^* \to \infty$ (region I in Fig. 2b). In addition, sensor cooperativity is less effective for less-correlated signals because a cooperative strategy relies on output suppression (which reduces both noise and output entropies). This strategy works well for more correlated signals as they carry less information (low signal entropy), which can be efficiently encoded by fewer output states. Thus, a reduction in noise entropy increases mutual information despite the decrease in output entropy. This is not the case for less-correlated signals, which carry more information (higher entropy) and which require more output states to encode effectively. As sensors become less noisy, the optimal strategy is nonequilibrium ($t^* \neq 0$; region III in Fig. 2b) only when the signal redundancy, i.e., the mutual information between the input signals $I(h_1; h_2)$, is relatively high. The sensing strategies $t = \pm|t^*|$ are time-reversed partners of one another both of which yield the same mutual information. This symmetry

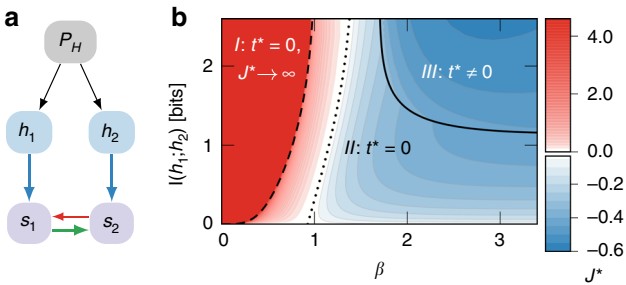

**Fig. 2 Optimal sensing for correlated signals.** Nonequilibrium sensing is optimal in the low-noise, high-correlation limit for correlated Gaussian signals. **a** We assume that the signal directly influences both sensors with varying correlation, Eq. (10). **b** The optimal coupling $J^*$ as a function of sensor reliability $\beta$ and the signal redundancy, i.e., the mutual information between the input signals $I(h_1; h_2)$. The optimal coupling diverges $J^* \to \infty$ at small $\beta$ (region I, left of the dashed curve) and decreases with larger $\beta$. Between the dashed and solid curves (region II), the mutual information $I(S; H)$ is maximised by equilibrium sensors with a finite $J^*$ that changes from cooperative (red) to anticooperative (blue) at the dotted line. Nonequilibrium sensing is the optimal strategy for signals with relatively high redundancy in the low-noise limit (region III, above the solid curve).

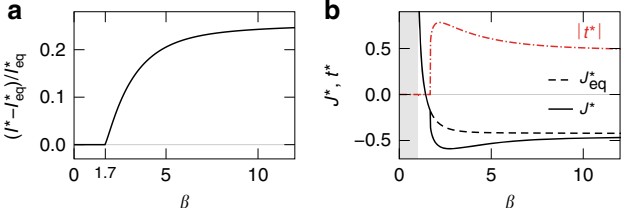

**Fig. 3 Optimal sensing in the high-signal-correlation limit.** For perfectly correlated Gaussian signals, the nonequilibrium information gain is largest in the noiseless limit. We consider a sensor complex driven with signal $H = (h, h)$ with $P_H = e^{-h^2/2}/\sqrt{2\pi}$. **a** The nonequilibrium gain as a function of sensor reliability $\beta$. The gain grows from zero at $\beta = 1.7$ and increases with $\beta$, suggesting that the enhancement results from the ability to distinguish additional signal features. **b** Optimal sensing strategy for varying noise levels. For $\beta < 1$ (shaded), the mutual information is maximised by an equilibrium system ($t^* = 0$) with infinitely strong coupling. The equilibrium strategy remains optimal for $\beta < 1.7$ with a coupling $J^*$ (solid) that decreases with $\beta$ and exhibits a sign change at $\beta = 1.4$. At bigger $\beta$, the optimal coupling in the equilibrium case (dashed) continues to decrease but equilibrium sensing becomes suboptimal. For $\beta > 1.7$, the mutual information is maximised by a finite nonequilibrium drive (dot-dashed) and negative coupling.

results from the fact that the signal prior $P_H$ (Eq. (10)) is invariant under $h_1 \leftrightarrow h_2$, hence the freedom in the choice of dominant sensor (i.e., we can either make $J_{12} > J_{21}$ or $J_{12} < J_{21}$).

Although Fig. 2b shows the results for positively correlated signals ($\alpha > 0$), the optimal sensing strategies for anticorrelated signals ($\alpha < 0$) exhibit the same dependence on sensor reliability and the signal redundancy but with the same nonequilibrium drive $t = \pm|t^*|$ and the optimal coupling $J^*$ that is opposite to that in the case $\alpha > 0$.

**Perfectly correlated signals**. To understand the mechanisms behind the optimal sensing strategy for correlated signals, we consider the limiting case of completely redundant Gaussian signals ($h_1 = h_2$), which captures most of the phenomenology depicted in Fig. 2b. We find that nonequilibrium drive allows further improvement on equilibrium sensors only for $\beta > 1.7$ and that the nonequilibrium gain remains finite as $\beta \to \infty$ (Fig. 3a). In Fig. 3b, we show the optimal parameters for both equilibrium and nonequilibrium sensing. The optimal coupling diverges for sensors with $\beta < 1$, decreases with increasing $\beta$ and exhibits a sign change at $\beta = 1.4$. For $\beta > 1.7$, the nonequilibrium drive is finite and the couplings are distinct $J^* \neq J^*_{eq}$.

Figure 4a compares the output and noise entropies of equilibrium and nonequilibrium sensing at optimal with that of noninteracting sensors for a representative $\beta = 4$. Here, anticooperativity ($J^*_{eq} < 0$) enhances mutual information in equilibrium sensing by maximising the output entropy, whereas nonequilibrium drive produces further improvement by lowering noise entropy. Compared with the noninteracting case Fig. 4b, optimal equilibrium sensing distributes the probability of the output states, $P_S$, more evenly (Fig. 4c), resulting in higher output entropy. This is because a negative coupling $J < 0$ favours the states $+-$ and $-+$, which are much less probable than $++$ and $--$ in a noninteracting system subject to perfectly correlated signals (Fig. 4b). However, this also leads to higher noise entropy, as the states $+-$ and $-+$ are equally likely for a given signal (Fig. 4c). By lifting the degeneracy between the states $+-$ and $-+$, nonequilibrium sensing suppresses noise entropy while maintaining a relatively even distribution of output states, Fig. 4d. As a result, for signals with low redundancy ($I(h_1; h_2) \lesssim 1$), a

nonequilibrium strategy allows no further improvement (Fig. 2b) because the states of a sensor complex are less likely to be degenerate (since the probability that $h_1 \approx h_2$ is smaller).

Although anticooperativity increases output entropy more than noise entropy at $\beta = 4$, it is not the optimal strategy for $\beta < 1.4$. For noisy sensors, a positive coupling $J > 0$ yields higher mutual information, Fig. 3b. This is because when the noise level is high, the output entropy is nearly saturated and an increase in mutual information must result primarily from the reduction of noise entropy by suppressing some output states–in this case, the states $+-$ and $-+$ are suppressed by $J > 0$.

We emphasise that the nonequilibrium gain is not merely a result of an additional sensor parameter. Instead of nonequilibrium drive we can introduce signal-independent biases on each sensor and keep the entire sensory complex in equilibrium. Such intrinsic biases can lower noise entropy in optimal sensing by breaking the degeneracy between the states $+-$ and $-+$ (Fig. 4b, c). However, favouring one sensor state over the other results in a greater decrease in output entropy and hence lower mutual information (see, Supplementary Fig. 1).

Our analysis does not rely on the specific Gaussian form of the prior distribution. Indeed, for correlated signals the nonequilibrium improvement in the low-noise, high-correlation limit is generic for most continuous priors (see, Supplementary Fig. 2).

**Time integration**. Cells do not generally have direct access to the receptor state. Instead, chemosensing relies on downstream readout molecules whose activation and decay couple to sensor states. Repeated interactions between receptors and a readout population provide a potential noise-reduction strategy through time averaging, which can compete with sensor cooperativity and energy consumption[14,16]. We generalise our model to incorporate time integration of the sensor state by a population of readout molecules and demonstrate that sensor coupling and nonequilibrium drive remains essential to optimal sensing.

We consider a system of binary sensors $S$, coupled to signals $H$ and a readout population $r$. We expand our original model (Nonequilibrium coupled sensors) to include readout activation ($r \to r+1$) and decay ($r \to r-1$), resulting in modified

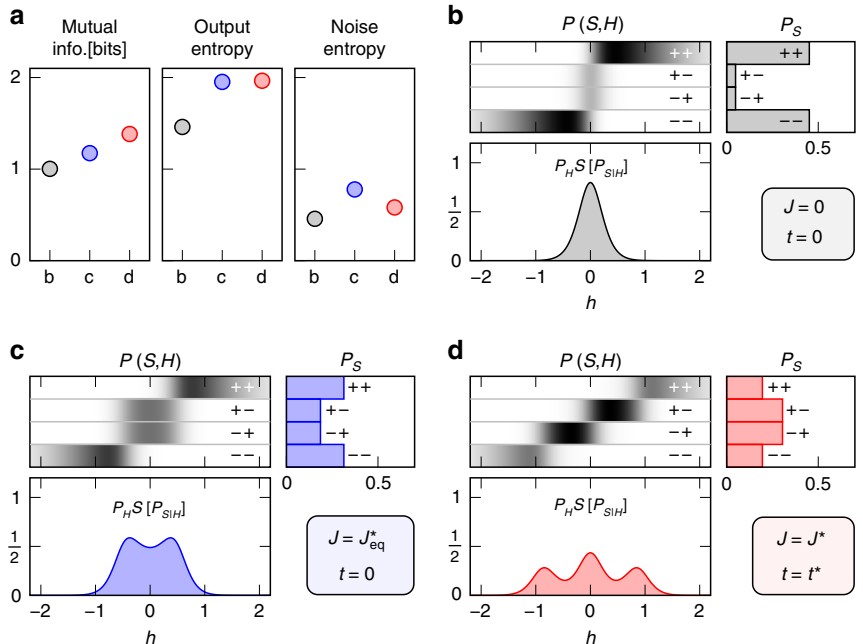

**Fig. 4 Mechanisms behind nonequilibrium performance improvement for joint, correlated signals.** For perfectly correlated Gaussian signals, in the low-noise limit, sensor anticooperativity ($J < 0$) increases the mutual information by maximising the output entropy while nonequilibrium drive ($t \neq 0$) provides further improvement by suppressing noise entropy. **a** We compare the mutual information, output, and noise entropies for three sensing strategies at $\beta = 4$: (**b** grey) noninteracting, (**c** blue) equilibrium and (**d** red) nonequilibrium cases. For each case, we find the configuration that maximises the mutual information under the respective constraints. **b–d** Signal-sensor joint probability distribution $P(S, H)$, output distribution $P_S$, and the "signal-resolved noise entropy" $P_H \mathcal{S}[P_{S|H}]$, corresponding to the cases shown in **a** where $\beta = 4$. Note that the noise entropy is the area under the signal-resolved curve. As the optimal coupling is negative ($J < 0$) at $\beta = 4$ (see Fig. 3b), the states $-+$ and $+-$, which are heavily suppressed by fully correlated signals in a noninteracting system (**b**), become more probable in the interacting cases, resulting in a more even output distribution **c**, **d** and thus a larger output entropy **a**. However anticooperativity also increases noise entropy in the equilibrium case **a** since the states $-+$ and $+-$ are degenerate **c**. By lifting this degeneracy (**d**), a nonequilibrium system can suppress the noise entropy and further increase mutual information **a**.

transition rates:

$$\Gamma_{S \to S^i|(r,H)} = \mathcal{N}_H e^{-\beta s_i \left( h_i - b_i + \sum_{j \neq i} J_{ij} s_j + \delta \mu_i r \right)} \quad (11)$$

$$\Gamma_{r \to r \pm 1|(S,H)} = \mathcal{N}_H e^{\pm \frac{1}{2} \beta \sum_i \delta \mu_i s_i} \quad (12)$$

where $S \to S^i$ denotes sensor inversion $s_i \to -s_i$, $r$ the readout population, $b_i$ the sensor-specific bias and $\mathcal{N}_H$ the overall timescale constant. We also introduce the sensor-dependent differential readout potential $\delta \mu_i$; the sensor state $s_i = \text{sgn}(\delta \mu_i)$ favours a larger readout population (by increasing activation rate and suppressing decays), whereas $s_i = -\text{sgn}(\delta \mu_i)$ biases the readout towards a smaller population. This allows the readout population to store samplings of sensor states over time, thus providing a physical mechanism for time integration of sensor states. The readout population in turn affects sensor inversions: the larger the readout population, the more favourable $s_i = \text{sgn}(\delta \mu_i)$ over $s_i = -\text{sgn}(\delta \mu_i)$. This two-way interplay between sensors and readouts is essential for a consistent equilibrium description, for one-way effects (e.g., sampling sensor states without altering the sensor complex) require Maxwell's demon—a nonequilibrium process not described by the model.

We further assume a finite readout population $r \leq r_0$, and that the readout activation and decay are intrinsically stochastic. Consequently, a readout population has a limited memory for past sensor states. Indeed, readout stochasticity sets a timescale beyond, which an increase in measurement time cannot improve sensing performance[6]. To investigate this fundamental limit, we let the measurement time be much longer than any stochastic

timescale. In this case, the sensor-readout joint distribution $P_{r,S|H}$ satisfies the steady state master equation with the transition rates in Eqs. (11), (12) (see, Steady state master equation in Methods).

When $J_{ij} = J_{ji}$, the steady state distribution obeys the detailed balance condition (see, Coupling symmetry and detailed balance in Methods) and is given by

$$P^{eq}_{r,S|H} = e^{-\beta F_{r,S|H}} / \mathcal{Z}^{eq}_H \quad (13)$$

with the free energy

$$F_{r,S|H} = -\sum_i (h_i - b_i) s_i - \sum_{i,j}^{j>i} J_{ij} s_i s_j - \sum_i \delta \mu_i s_i r \quad (14)$$

This distribution results in $P^{eq}_{r|S,H} = P^{eq}_{r|S}$, implying a Markov chain $H \to S \to r$ ($H$ affects $r$ only via $S$), hence the data processing inequality $I(S; H) \geq I(r; H)$. That is, in equilibrium, time integration of sensor states cannot produce a readout population that contain more information about signals than the sensor states[7] (see, also refs. [18,19]). This result applies to any equilibrium sensing complexes (see, Equilibrium time integration in Methods).

We now specialise to the case of two coupled sensors $S = (s_1, s_2)$ and introduce two new variables, $\Delta$ and $\delta$, defined via

$$\delta \mu_1 = \frac{1}{2}(\Delta + \delta) \quad \text{and} \quad \delta \mu_2 = \frac{1}{2}(\Delta - \delta) \quad (15)$$

For this parametrisation, the effective chemical potentials for readout molecules are given by

$$\mu_{++} = \Delta, \quad \mu_{--} = -\Delta, \quad \mu_{+-} = \delta, \quad \mu_{-+} = -\delta,$$

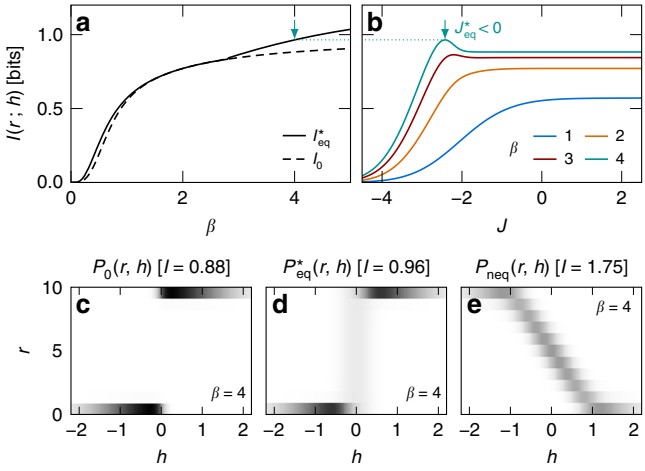

**Fig. 5 Optimal sensing in the presence of time integration.** Sensor interaction and energy consumption remain important for optimal sensing in the presence of time integration of sensor states by a population of readout molecules. We consider a two-sensor complex, driven by signal $H = (h, h)$ with $P_H = e^{-h^2/2}/\sqrt{2\pi}$ and coupled to a readout population $r$ (Eqs. (11), (12)). **a** Mutual information between the signal and the readout population for noninteracting sensors (dashed) and optimal equilibrium sensors (solid). Independent sensors are always suboptimal. **b** Mutual information as a function of sensor coupling $J$ at various sensor reliabilities $\beta$ (see legend). The optimal equilibrium strategy changes from cooperativity ($J > 0$) in the noisy limit to anticooperativity ($J < 0$) in the low-noise limit. **c-e** Joint probability distribution of signal and readout population at $\beta = 4$ for three sensing systems: noninteracting **c**, optimal equilibrium (**d**), and near-optimal nonequilibrium (**e**) (cf. Fig. 4). Optimal equilibrium sensors use anticooperativity to increase the probability of the states $+-$ and $-+$, which map to intermediate readout populations $0 < r < r_0$, and as a result, allows for a more efficient use of output states compared with noninteracting sensors. Nonequilibrium drive lifts the degeneracy of intermediate readout states, leading to an even more effective use of readout states. For the nonequilibrium example in **e**, we obtain $I_{neq} = 1.75$ bits, compared with $I_{eq}^* = 0.96$ bits for optimal equilibrium sensors at the same $\beta$. In **a-e**, we use $\Delta = 1$ and $r_0 = 10$, and in **e**, $J = -2$, $t = 7$, and $\delta = -0.6$.

where $\mu_S = \Sigma_i \delta\mu_i s_i$. We see that $\Delta$ ($\delta$) parametrises how different the sensor states $++$ and $--$ ($+-$ and $-+$) appear to the readout population. To maximise utilisation of readout states, we set the sensors biases to $b_i = \delta\mu_i r_0/2$ where $r_0$ is the maximum readout population. For optimal equilibrium sensing, we maximise mutual information $I(r; H)$ by varying $J$ and $\delta$ under the constraint $t = 0$, and the noninteracting case corresponds to $J = t = \delta = 0$.

Figure 5a depicts the readout-signal mutual information for perfectly correlated Gaussian signals for the cases of optimal equilibrium sensing (solid) and noninteracting sensors (dashed). Independent sensors are suboptimal for all $\beta$, i.e., we can always increase mutual information by tuning $\delta$ and $J$ away from zero. At maximum mutual information we find $\delta = 0$ and $J \neq 0$. In Fig. 5b, we show the mutual information as a function of $J$ (at $\delta = 0$). We see that $J_{eq}^* \to \infty$ in the noisy limit (low $\beta$) and $J_{eq}^* < 0$ in the low-noise limit (high $\beta$). This crossover from cooperativity to anticooperativity is consistent with our results in Correlated signals and Perfectly correlated signals (see also, Figs. 2 and 3b).

To reveal the mechanism behind optimal equilibrium sensing in the low-noise limit, we examine the joint probability distribution $P(r, h)$ at $\beta = 4$ for noninteracting (Fig. 5c) and optimal equilibrium sensors (Fig. 5d). We see that anticooperativity increases mutual information by distributing the output (readout)

states more efficiently (cf. Fig. 4b, c). Noninteracting sensors partition outputs into large and small readout populations, which corresponds to positive and negative signals, respectively. This is because correlated signals favour the chemical potentials $\mu_{++} = -\mu_{--} = \Delta$, which bias the readout population towards $r = 0$ and $r = r_0$, and suppress $\mu_{+-} = -\mu_{-+} = \delta = 0$, which encourage evenly distributed readout states. By adopting an anticooperative strategy ($J < 0$) to counter signal correlation, optimal equilibrium sensors can use more output states (on average) to encode the signal. The increase in accessible readout states also raises noise entropy, but the increase in output entropy dominates in the low-noise limit, resulting in higher mutual information.

Finally, we demonstrate that energy consumption can further enhance sensing performance. Figure 5e shows $P(r, h)$ for a nonequilibrium sensor complex. We see that nonequilibrium drive lifts the degeneracy in intermediate readout states ($0 < r < r_0$), leading to a much more effective use of output states. For the nonequilibrium sensor complex in Fig. 5e, we obtain $I_{neq}(r; h) = 1.75$ bits, compared with $I_{eq}^*(r; h) = 0.96$ bits for optimal equilibrium sensors (Fig. 5a, b) at the same sensor reliability ($\beta = 4$). We note that this nonequilibrium gain relies also on $\delta \neq 0$ to distinguish the sensor states $+-$ and $-+$. The staircase of readout states in Fig. 5e corresponds to the anticorrelated sensor states $+-$ and $-+$ which do not always favour higher readouts at positive signals (see also, Supplementary Figs. 3, 4).

## Discussion

We introduce a minimal model of a sensor complex that encapsulates both sensor interactions and energy consumption. For correlated signals, we find that sensor interactions can increase sensing performance of two binary sensors, as measured by the steady state mutual information between the signal and the states of the sensor complex.

This result highlights sensor cooperativity as a biologically plausible sensing strategy[11–13]. However, the nature of the optimal sensor coupling does not always reflect the correlation in the signal; for positively correlated signals, the optimal sensing strategy changes from cooperativity to anticooperativity as the noise level decreases, see also ref. [17]. Anticooperativity emerges as the optimal strategy through countering the redundancy in correlated signals by suppressing correlated outputs, and thus redistributing the output states more evenly. The same principle also applies to population coding in neural networks[17], positional information coding by the gap genes in the *Drosophila* embryo[20–22] and time-telling from multiple readout genes[23]. Surprisingly, we find that energy consumption leads to further improvement only when the noise level is low and the signal redundancy high.

We find that sensor coupling and energy consumption remain important for optimal sensing under time integration of the sensor state—a result contrary to earlier findings that a cooperative strategy is suboptimal even when sensor interaction can couple to nonequilibrium drive[14,16]. This discrepancy results from an assumption of continuous, deterministic time integration that requires an infinite supply of readout molecules and external nonequilibrium processes, and which also leads to an underestimation of noise in the output; we make no such assumption in our model. In addition, we use the data processing inequality to show for any sensing system that time integration cannot improve sensing performance unless energy consumption is allowed either in sensor coupling or downstream networks (see also refs. [5,7]).

Our work highlights the role of signal statistics in the context of optimal sensing. We show that a signal prior distribution is an important factor in determining the optimal sensing strategy as it sets the amount of information carried by a signal. With a signal

prior, we quantify sensing performance by mutual information, which is a generalisation of linear approximations used in previous works[6–8,11,14,16].

To focus on the effects of nonequilibrium sensor cooperation, we neglect the possibility of signal crosstalk and the presence of false signals. Limited sensor-signal specificity places an additional constraint on sensing performance[24] (but see, ref. [25]). Previous works have shown that kinetic proofreading schemes[3] can mitigate this problem for isolated chemoreceptors that bind to correct and incorrect ligands[26,27]. Our model can be easily generalised to include crosstalk, and it would be interesting to investigate whether nonequilibrium sensor coupling can provide a way to alleviate the problem of limited specificity.

Although we considered a simple model, our approach provides a general framework for understanding collective sensing strategies across different biological systems from chemoreceptors to transcriptional regulation to a group of animals in search of mates or food. In particular, possible future investigations include the mechanisms behind collective sensing strategies in more complex, realistic models, non-binary sensors, adaptation, and generalisation to a larger number of sensors. It would also be interesting to study the channel capacity in the parameter spaces of both the sensor couplings and the signal prior, an approach that has already led to major advances in the understanding of gene regulatory networks[28]. Finally, the existence of optimal collective sensing strategies necessitates a characterisation of the learning rules that gives rise to such strategies.

## Methods

**Steady state master equation**. The steady state probability distribution satisfies a linear matrix equation

$$\sum_j W_{ij} p_j = 0 \tag{16}$$

where $p_j$ denotes the probability of the state $j$. The matrix $W$ is defined such that

$$W_{ij} = \Gamma_{j \to i} \text{ for } i \neq j \quad \text{and} \quad W_{kk} = -\sum_i \Gamma_{k \to i} \tag{17}$$

where $\Gamma_{j \to i}$ denotes the transition rate from the states $j$ to $i$ and $\Gamma_{j \to j} = 0$. The solution of Eq. (16) corresponds to a direction in the null space of the linear operator $W$. For two coupled sensors considered in Nonequilibrium coupled sensors, Eq. (16) (Eq. (2)) becomes a set of four simultaneous equations, which we solve analytically for a solution (Eq. (7)) that also satisfies the constraints of a probability distribution ($\sum_j p_j = 1$, $p_i \in \mathbb{R}$ and $p_i \geq 0$ for all $i$). In a larger system an analytical solution is not practical. For example, in 'Time integration' we consider a 44-state system of two sensors with at most 10 readout molecules. In this case we obtain the null space of the matrix $W$ from its singular value decomposition, which can be computed with a standard numerical software.

**Coupling symmetry and detailed balance**. Here we show that the transition rates in Eq. (1) does not satisfy Kolmogorov's criterion—a necessary and sufficient condition for detailed balance—unless $J_{ij} = J_{ji}$. Consider the sensors $s_i$ and $s_j$ in a sensor complex $S = \{s_1, s_2, \ldots, s_N\}$ with $N > 1$. This sensor pair admits four states $(s_i, s_j) = --, -+, ++, +-$. The transitions between these states form two closed sequences in opposite directions

$$
\begin{array}{ccc}
-- & \underset{\Gamma_{a'}}{\overset{\Gamma_a}{\rightleftarrows}} & -+ \\
\Gamma_d \big\updownarrow \Gamma_{d'} & & \Gamma_{b'} \big\updownarrow \Gamma_b \\
+- & \underset{\Gamma_c}{\overset{\Gamma_{c'}}{\rightleftarrows}} & ++
\end{array}
\tag{18}
$$

Kolmogorov's criterion requires that, for any closed loop, the product of all transition rates in one direction must be equal to the product of all transition rates in the opposite direction—i.e., $\Gamma_a \Gamma_b \Gamma_c \Gamma_d = \Gamma_{a'} \Gamma_{b'} \Gamma_{c'} \Gamma_{d'}$. For the rates in Eq. (1), we have

$$\frac{\Gamma_a \Gamma_b \Gamma_c \Gamma_d}{\Gamma_{a'} \Gamma_{b'} \Gamma_{c'} \Gamma_{d'}} = e^{4\beta(J_{ij} - J_{ji})} \tag{19}$$

therefore, only symmetric interactions satisfy Kolmogorov's criterion and any

asymmetry in sensor coupling necessarily breaks detailed balance. This result holds also for the generalised transition rates in Eqs. (11), (12).

**Equilibrium time integration**. Following the analysis in Supplemental Material of ref. [7], we provide a general proof that equilibrium time integration of receptor states cannot generate readout populations that contain more information about the signals than the receptors. We consider a system of receptors $S = (s_1, s_2, \ldots, s_N)$, driven by signal $H = (h_1, h_2, \ldots, h_N)$ and coupled to readout populations $R = (r_1, r_2, \ldots, r_M)$. In equilibrium, this system is described by a free energy

$$F_{R,S|H} = f(H, S) + g(S, R) \tag{20}$$

where $f(H, S)$ and $g(S, R)$ describe signal-sensor and sensor-readout couplings, respectively, and include interactions among sensors and readout molecules. The Boltzmann distribution for sensors and readouts reads

$$P_{R,S|H} = e^{-\beta[f(H,S) + g(S,R)]} / Z_H \tag{21}$$

with the partition function $Z_H$. Therefore, we have

$$P_{R|S,H} = \frac{P_{R,S|H}}{\sum_R P_{R,S|H}} = \frac{e^{-\beta g(S,R)}}{\sum_R e^{-\beta g(S,R)}} = P_{R|S} \tag{22}$$

where the summation is over all readout states. This allows the decomposition of the joint distribution, $P_{R,S|H} = P_{R|S,H} P_{S|H} = P_{R|S} P_{S|H}$, which implies a Markov chain $H \to S \to R$—that is, $H$ affects $R$ only through $S$. (This does not mean $R$ does not affect $S$, for $P_{S|R,H}$ still depends on $R$.) From the data processing inequality, it immediately follows that $I(H; S) \geq I(H; R)$. We emphasise that this constraint applies to any equilibrium sensor complex and downstream networks, which can be described by the free energy in Eq. (20), regardless of the numbers of sensors and readout species, sensor characteristics (e.g., number of states), sensor-signal and sensor-readout couplings (including crosstalk), and interactions among sensors and readout molecules.

## Data availability

Data sharing not applicable to this article as no datasets were generated or analysed during the current study.

## Code availability

A Mathematica code for computing the optimal sensor parameters $(J^*, t^*)$ that maximise the mutual information between two coupled sensors and correlated Gaussian signals is available at https://github.com/vn232/NeqCoopSensing.

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

## Acknowledgements

We are grateful to Gašper Tkačik and Pieter Rein ten Wolde for useful comments and a critical reading of the manuscript. V.N. acknowledges support from the National Science Foundation under Grants DMR-1508730 and PHY-1734332, and the Northwestern-Fermilab Center for Applied Physics and Superconducting Technologies. G.J.S. acknowledges research funds from Vrije Universiteit Amsterdam and OIST Graduate University. D.J.S. was supported by the National Science Foundation through the Center for the Physics of Biological Function (PHY-1734030) and by a Simons Foundation fellowship for the MMLS. This work was partially supported by the National Institutes of Health under award number R01EB026943 (V.N. and D.J.S.).

## Author contributions

V.N., D.J.S., and G.J.S. conceived the study, interpreted the results and wrote the manuscript. V.N. performed numerical calculations.

## Competing interests

The authors declare no competing interests.
