## [Peer Review File · Nature Communications]

Reviewers' comments:

Reviewer #1 (Remarks to the Author):

What are the major claims of the paper? Are they novel and will they be of interest to others in the community and the wider field? If the conclusions are not original, it would be helpful if you could provide relevant references. Is the work convincing, and if not, what further evidence would be required to strengthen the conclusions? On a more subjective note, do you feel that the paper will influence thinking in the field? Please feel free to raise any further questions and concerns about the paper.

We would also be grateful if you could comment on the appropriateness and validity of any statistical analysis, as well the ability of a researcher to reproduce the work, given the level of detail provided.

“Energy consumption and cooperation for optimal sensing”

Vudtiwat Ngampruetikorn, David J. Schwab, and Greg J. Stephens

In this work the authors introduce a kinetic model of multiple coupled sensors (modeling e.g. receptors) detecting correlated signals (modeling e.g. ligand concentrations). For the special case of two sensors and two signals, they vary the (in general, out of equilibrium) sensor coupling to optimize information transmission from fields to the joint state of the sensors, as a function of intrinsic noise and signal correlation. In the main text and SI, they explore the optimal information transmission, and the advantages accrued from sensor coupling and from nonequilibrium drive, for correlated bivariate Gaussian signals, perfectly correlated unimodal or bimodal signals, and finally a single Gaussian or Bernoulli signal. They place particular emphasis on their findings that nonequilibrium drive only enhances information transmission for highly correlated signals and low intrinsic noise.

I like this paper. It is generally well written, with relatively clear flow and accessible language and figures. The special cases and the supplementary material do a good job of walking the reader through the intuitions underlying their major findings, which are convincing within the context of this particular model, though their generality could perhaps be stated more carefully (I detail further below). My sense is that this study of the interactions between signal correlations, sensor cooperativity, and nonequilibrium kinetics represents an important step beyond previous work; though as an interested observer but not an experienced insider in this sub-field, I would have benefited from more explicit statement of what the authors take to be the critical advances distinct from the previous literature.

I find the subject matter of this paper interesting: there is a general sense that energy consumption, and the nonequilibrium kinetics it enables, permits greater performance in a variety of contexts, in sensing but also kinetic proofreading and applications of the thermodynamic uncertainty relation; this paper gives a concrete but presumably quite generally applicable sensing model that demonstrates with finer detail just what kinds of circumstances can benefit from nonequilibrium kinetics. I expect that this paper will be of interest to statistical physicists, systems biologists, and biophysicists, and I do expect it to influence thinking about the role of nonequilibrium kinetics in biological function.

Thus I recommend acceptance of this manuscript. But I only came to this conclusion after spending significant time working through the manuscript, so I think there is significant room to improve its presentation. I give details for these important points below.

BUILDING INTUITION

* p3 mid right: "sensor cooperativity is less effective for less correlated signals because while more information is encoded in a less correlated prior, the information capacity of a sensor complex decreases as a cooperative strategy relies on output suppression (which reduces both noise and output entropies)". How is information encoded in a prior? What do you mean by "information capacity"? Finally, reduced noise and output entropies could either increase or decrease information, depending on the relative magnitude of their changes, so the end of this sentence needs clarification.

* p3 bottom right: By "opposite to those in the case $\alpha > 0$ ", do you mean that J^* and t^* are the negative of their values for positive correlations? If so, why would the optimal t^* flip sign when the correlations are negative? Similarly, in Fig S1, what leads to the asymmetry that makes negative t interact well with positive J , and positive t interact well with negative J ? You later interpret the sign of t as indicating the relative strength of influence of each sensor on the other, which is intuitively important when one sensor has direct access to ligand concentrations and the other does not. But I don't understand why changing the sign of the correlation, or the sign of the sensor coupling, would lead you to want to change the sign of t .

REPRODUCIBILITY

p2 mid right: Give further argument/citation for why you need symmetric J 's for detailed balance, specifically in the case of more than 2 sensors.
Give more argument for how you get Eq. 7 from Eq. 2.
Describe how you found the optimal J^* and t^* .

CLARIFY

Eq. 1 and subsequent text: Eq. 1 looks like a statement of the equilibrium probability for an Ising model, not a rate. Say anything about why you choose this form, to help orient the reader? What is a "normalisation" of a rate? Be more precise than that N_H "drops out in steady state".
Fig 3 caption: "The nonequilibrium information gain is largest in the noiseless limit for perfectly correlated Gaussian signals." Not sure how to parse this.
pS1 top left: "Univariate Signals" doesn't really capture the relevant distinction here, namely that one sensor doesn't directly detect any signal. Suggest renaming the title and use of this phrase in text.
pS2 top right: "our results hold also for discrete priors" 1) All the priors in Fig S5 are continuous. 2) You haven't shown a general result, you've shown a qualitatively consistent set of results for 4 different priors.
pS2 top right: don't know what "vacant, occupied states and binary conformal states" means.

DATA DISPLAY

Fig 3 caption: c): Can't see on graph what information enhancement is for $\beta < 1.7$.
Fig S1: I/I_0 here, but $(I-I_0)/I_0$ elsewhere. Be consistent unless there is some overriding reason to display it differently in different plots.
pS1 mid left: "optimal sensing corresponds to an infinite J and a finite t (Fig. S1d)". Fig only shows t , not diverging J .
Fig S5 caption: "clearly finite": I wouldn't say it is clear, as there certainly appears to be some blue at/near $J=t=0$.

MANUSCRIPT ORGANIZATION

Most of SI is not explicitly referenced in the main text.
Why doesn't Fig S4 come next after Fig S1, since it is referenced next after S1?
Fig S3 is not described in the SI text.

POSSIBLE TYPOS:

p1 bottom left: "... influence sensitivity [COMMA] and receptor cooperativity ..."

p1 middle right: comma should be period.

Eq. 1 and Eq. 4 appear to have opposite sign conventions.

p2 top right: " $\{h_i\}$ parametrises the concentration change": this should be concentration, not concentration change, right?

Fig. 3 caption: "continues to decreases"

Fig. 4: some labels are cut off

pS1, mid left: "... that an improvement on equilibrium sensing at all reliability ..."

Fig S4 caption: flipped solid and dashed lines

pS2 top right: "AND our results hold ..."

Reviewer #3 (Remarks to the Author):

The manuscript by Ngampruetikorn et al. investigates the roles of cooperativity and energy consumption in noisy signal detection using a minimal coupled-receptor model. They find that when the input signals impinging on each receptor are uncorrelated, the receptors should be uncoupled. When the inputs are correlated, the optimal receptor coupling depends on the input noise: a noisy input is best detected by cooperative receptors because cooperativity reduces noise, whereas a reliable input is best detected by anti-cooperative receptors because anti-cooperativity optimally utilizes all receptor output states. Finally, if the correlated inputs are sufficiently reliable, the optimal anti-cooperative receptor strategy becomes dissipative because dissipation lifts the redundancy between the +- and -+ anti-correlated output states.

The study is well conceived and well explained. The model is minimal enough to be analytically tractable, yet general enough to include noise, cooperativity, and nonequilibrium driving as tunable parameters. The results are clear, and the mechanisms behind them are reasonably well elucidated.

The thing that I am struggling with, then, is how to place these results in the larger context of the field. Thermodynamics of sensing is an important topic of recent interest, and the authors do a good job of reviewing the recent literature. However, they do not circle back and make clear whether their results agree with, contradict, or go beyond any of these recent results. Consequently, it is difficult to gauge the impact of their study. Has it changed the perspective of how we should think of the thermodynamics of sensing in any way? There needs to be a clear answer to this question if this study is to be of interest to the readers of this journal.

Specifically:

1. The authors cite works demonstrating that cooperativity suppresses noise in steady state. The present results are in steady state and therefore consistent with these works, correct?
2. The authors also cite works demonstrating that non-cooperative receptors are optimal when there is an integration time. The present work does not consider an integration time and therefore does not contradict these works, correct? But integrating a signal over time is a very natural scenario. Why would the authors not look at a time-integrated output in their model? It seems like this might make cooperativity always suboptimal and therefore change the main results of the study.
3. Finally, the authors cite works that derive the tradeoffs for noise reduction among energy consumption, integration time, and molecule number. Again, these works seem to have a partial overlap with the authors' present focus on energy consumption and cooperativity, but these works also address time integration in a natural way. How do the authors' results relate to these other

works and therefore advance the field?

Other comments:

4. Fig 1c is mentioned first. Why not switch 1c with 1a and b?

5. Fig 2b contains most of the main results of the study, but it took me a long time to fully figure it out. Part of the reason is that there is information in the figure that is not clear from the text, and vice versa. To help, the authors might consider adding information to the I, II, and III labels, e.g., 'I: $J^* \rightarrow \infty, t^* = 0$ ', 'II: $t^* = 0$ ', 'III: $t^* \neq 0$ '. Similarly, refer to regions I, II, and III in the text.

6. Another thing that I struggled with in Fig 2b is why $I(h_1, h_2)$ is used as an axis. This quantity is not mentioned until the end of the text discussing Fig 2b. Why not just use α for this axis? It would also prevent confusion from using two mutual-information quantities in the study.

7. Fig 3a and b are never mentioned in the text. Are they necessary?

8. Fig S1, S2, and S4 are never mentioned in the text. Again, are they necessary?

Summary of changes

All changes are blue in the revised manuscript.

Major changes:

1. We added a new subsection *Time integration* (including Fig. 5) in the *Results* section. Here we investigated the effects of time integration of the sensor state on optimal sensing strategies.
2. We added two new paragraphs in *Discussion* to summarise our results on time integration of sensor states and discuss our work in relation to existing literature.
3. We added the *Methods* section containing three subsections — i. *Steady state master equation*, ii. *Coupling symmetry and detailed balance* and iii. *Equilibrium time integration* — to which we refer throughout the revised manuscript.
4. We removed the *Univariate Signals* section (including Figs. S1, S2 & S4) from *Supplementary Information* since it is not essential to understanding of our main results.

Minor changes:

5. We have improved the clarity of the presentation throughout the revised manuscript.
6. We added a sentence towards the end of *Abstract* and *Introduction* to mention our analysis of time integration.
7. We edited the text around Eq. 1 to provide the intuition behind each term and explain why the constant prefactor drops out in steady state.
8. We now provide an example code for computing the optimal sensor parameters J^* and t^* in the *Code Availability* section.
9. We edited the text in *Correlated signals* to clarify our explanation of why cooperativity is less effective for less correlated signals.
10. We moved our remarks on non-Gaussian priors from *Discussion* to the end of *Perfectly correlated signals* in *Results*.
11. We added a sentence in the first paragraph of *Discussion* to place our results on sensor cooperativity in the context of previous works.
12. We added a paragraph at the end of *Correlated signals* to explain why the optimal strategy (J^*, t^*) changes sign as the signal correlation is inverted.
13. We rearranged the panels in Fig. 1, as suggested by Reviewer 3.
14. We added more descriptive labels in Fig. 2b and referred to them in the text.
15. We modified Fig. 3c to illustrate more clearly the region $\beta < 1.7$, and we edited the first sentence in the caption for clarity.

Response to Reviewer 1

We thank the reviewer for a thorough reading of our work, and we are pleased that they recommend acceptance of our manuscript. Below we address the reviewer's comments point by point.

[...] though as an interested observer but not an experienced insider in this sub-field, I would have benefited from more explicit statement of what the authors take to be the critical advances distinct from the previous literature.

To better place our work in the context of existing literature, we expanded our manuscript to provide an analysis of the effects of time integration of sensor states on optimal sensing (*Change 1*), and we added two paragraphs in *Discussion* to contrast our results with the conclusions from previous works as well as to highlight the importance of signal statistics in the context of chemical sensing (*Change 2*).

BUILDING INTUITION

p3 mid right: "sensor cooperativity is less effective for less correlated signals because while more information is encoded in a less correlated prior, the information capacity of a sensor complex decreases as a cooperative strategy relies on output suppression (which reduces both noise and output entropies)". How is information encoded in a prior? What do you mean by "information capacity"? Finally, reduced noise and output entropies could either increase or decrease information, depending on the relative magnitude of their changes, so the end of this sentence needs clarification.

We have edited this sentence for clarity (*Change 9*).

p3 bottom right: By "opposite to those in the case $\alpha > 0$ ", do you mean that J^* and t^* are the negative of their values for positive correlations? If so, why would the optimal t^* flip sign when the correlations are negative? [...] But I don't understand why changing the sign of the correlation, or the sign of the sensor coupling, would lead you to want to change the sign of t .

This behaviour results from the fact that the mutual information is invariant under $(\alpha, J, t) \rightarrow (-\alpha, -J, -t)$ which in turn follows from the fact that $(\alpha, J, t, s_2, h_2) \rightarrow (-\alpha, -J, -t, -s_2, -h_2)$ leaves the distributions $P_{S|H}$ and P_H unchanged (see, Eqs. 4, 7, 8 & 10). Note that S and H are integrated out when we compute the mutual information. We have included this explanation in the revised manuscript (*Change 12*).

REPRODUCIBILITY

p2 mid right: Give further argument/citation for why you need symmetric J 's for detailed balance, specifically in the case of more than 2 sensors.

We have added a reference to the new subsection in *Methods* which provides a proof that asymmetric coupling always breaks detailed balance in sensing systems with two or more sensors (*Change 3*).

Give more argument for how you get Eq. 7 from Eq. 2.

We have included a subsection in the *Methods* section to discuss the general structure of steady state master equations and the methods for obtaining the solutions (*Change 3*).

Describe how you found the optimal J^* and t^* .

We now provide an example code for computing the optimal sensor parameters J^* and t^* in the *Code Availability* section (*Change 8*).

CLARIFY

Eq. 1 and subsequent text: Eq. 1 looks like a statement of the equilibrium probability for an Ising model, not a rate. Say anything about why you choose this form, to help orient the reader?

We now explained the intuition behind each term in the transition rate in the sentences following Eq. 1 (*Change 7*).

What is a “normalisation” of a rate? Be more precise than that N_H “drops out in steady state”.

We agree that ‘normalisation’ is a misnomer which we now replaced with ‘constant’, and we added a brief explanation as to why N_H drops out in steady state (*Change 7*).

Fig 3 caption: “The nonequilibrium information gain is largest in the noiseless limit for perfectly correlated Gaussian signals.” Not sure how to parse this.

We have edited this sentence for clarity (*Change 15*).

- pS1 top left: “Univariate Signals” doesn’t really capture the relevant distinction here, namely that one sensor doesn’t directly detect any signal. Suggest renaming the title and use of this phrase in text.
- a** pS2 top right: “our results hold also for discrete priors” 1) All the priors in Fig S5 are continuous. 2) You haven’t shown a general result, you’ve shown a qualitatively consistent set of results for 4 different priors.
- b** pS2 top right: don’t know what “vacant, occupied states and binary conformal states” means.

We have removed this section (*Change 4*), but for transparency we respond to the reviewer’s comments on the content of this section:

- a** This is a typo. We should have referred to Fig. S4c (instead of Fig. S5) which depicts the results for discrete binary signals.
- b** In the four-state receptor model, each receptor has two conformal states and can be either vacant or occupied (i.e., whether it is bound to a ligand). Two binary sensors are

equivalent to one four-state receptor when we identify the states of one binary sensor with the vacant and occupied states, and the states of the other binary sensor with the two conformal states.

DATA DISPLAY

Fig 3 caption: c): Can't see on graph what information enhancement is for $\beta < 1.7$.

We have improved the visibility in the region $\beta < 1.7$ (*Change 15*).

- Fig S1: I/I_0 here, but $(I - I_0)/I_0$ elsewhere. Be consistent unless there is some overriding reason to display it differently in different plots.
- pS1 mid left: “optimal sensing corresponds to an infinite J and a finite t (Fig. S1d)”. Fig only shows t, not diverging J.
- Fig S5 caption: “clearly finite”: I wouldn't say it is clear, as there certainly appears to be some blue at/near $J=t=0$.

MANUSCRIPT ORGANIZATION

- Most of SI is not explicitly referenced in the main text.
- Why doesn't Fig S4 come next after Fig S1, since it is referenced next after S1?

We have removed this section (*Change 4*).

Fig S3 is not described in the SI text.

We refer to this figure from the main text in the penultimate paragraph of *Perfectly correlated signals* in *Results*. Note that this figure becomes Supplementary Figure 1 in the revised manuscript.

POSSIBLE TYPOS

- p1 bottom left: “. . . influence sensitivity [COMMA] and receptor cooperativity . . .”
- p1 middle right: comma should be period.
- a** Eq. 1 and Eq. 4 appear to have opposite sign conventions.
- b** p2 top right: “ $\{h_i\}$ parametrises the concentration change”: this should be concentration, not concentration change, right?
- Fig. 3 caption: “continues to decreases”
- Fig. 4: some labels are cut off

We thank the reviewer for spotting these typos. We have corrected all of these in the revised manuscript with two exceptions (**a** and **b** above) which we elaborate below.

a Eqs. 1&4 are consistent. From Eq. 5, we see that the transition rate out of a state S (Eq. 1) is proportional to $e^{\beta F_S}$ (the *inverse* of the Boltzmann factor) with the free energy F_S defined in Eq. 4. This means the out-transition rate is greater (shorter lifetime) for a state with higher free energy.

b Chemoreceptors respond to the changes in chemical concentration with respect to the background levels, and we define the signals $\{h_i\}$ to reflect this.

- pS1, mid left: “. . . that an improvement on equilibrium sensing at all reliability . . . ”
- Fig S4 caption: flipped solid and dashed lines
- pS2 top right: “AND our results hold . . . ”

We have removed this section (Change 4).

Response to Reviewer 3

We thank the reviewer for a careful reading of our manuscript. We are pleased that the reviewer finds our work ‘*well conceived*’ and ‘*well explained*’, and our results ‘*clear*’ and ‘*well elucidated*.’ Below we respond to the reviewer’s suggestions point by point.

[. . .] However, they do not circle back and make clear whether their results agree with, contradict, or go beyond any of these recent results. Consequently, it is difficult to gauge the impact of their study. Has it changed the perspective of how we should think of the thermodynamics of sensing in any way? [. . .]

In two new paragraphs in *Discussion*, we discuss our model and results in relation to previous works, and we highlight the importance of signal statistics in the context of optimal sensing (Change 2). To make a stronger connection to existing literature, we also included a new analysis of time integration of sensor states (Change 1).

1. The authors cite works demonstrating that cooperativity suppresses noise in steady state. The present results are in steady state and therefore consistent with these works, correct?

We added a sentence in *Discussion* to place our results on cooperativity in relation to these previous works (Change 11).

2. The authors also cite works demonstrating that non-cooperative receptors are optimal when there is an integration time. The present work does not consider an integration time and therefore does not contradict these works, correct? But integrating a signal over time is a very natural scenario. Why would the authors not look at a time-integrated output in their model? It seems like this might make cooperativity always suboptimal and therefore change the main results of the study.

We have investigated the effects of time integration of sensor states and presented the analyses and results in a new subsection (see, Change 1). By generalising our original model, we showed that cooperativity (as well as anticooperativity and energy consumption) remains an important strategy in optimal sensing, thus contradicting the conclusions

of previous works. We argued that the conclusions that independent receptors are optimal were based on unphysical assumptions in the model (see, Change 2).

3. Finally, the authors cite works that derive the tradeoffs for noise reduction among energy consumption, integration time, and molecule number. Again, these works seem to have a partial overlap with the authors' present focus on energy consumption and cooperativity, but these works also address time integration in a natural way. How do the authors' results relate to these other works and therefore advance the field?

We have now addressed time integration of sensor states in the revised manuscript, and we showed that our results do not change qualitatively (Change 1). Therefore our results offer a fresh insight into the role of signal statistics in the context of chemical sensing, and our manuscript has direct implications for future work seeking to quantify the physical limits of a sensor complex, evolved under a signal prior (Change 2).

4. Fig 1c is mentioned first. Why not switch 1c with 1a and b?

We have rearranged the panels in Fig. 1, as suggested (Change 13).

5. Fig 2b contains most of the main results of the study, but it took me a long time to fully figure it out. Part of the reason is that there is information in the figure that is not clear from the text, and vice versa. To help, the authors might consider adding information to the I, II, and III labels, e.g., 'I: $J^* \rightarrow \infty, t^* = 0$ ', 'II: $t^* = 0$ ', 'III: $t^* \neq 0$ '. Similarly, refer to regions I, II, and III in the text.

We have added descriptive labels and referred to them in the text (Change 14).

6. Another thing that I struggled with in Fig 2b is why $I(h_1, h_2)$ is used as an axis. This quantity is not mentioned until the end of the text discussing Fig 2b. Why not just use alpha for this axis? It would also prevent confusion from using two mutual-information quantities in the study.

We understand the reviewer's concern. However we believe that it is more natural to parametrise signal correlation with $I(h_1; h_2)$ since it directly measures the reduction in information content (entropy) of correlated signals.

7. Fig 3a and b are never mentioned in the text. Are they necessary?

Although Fig. 3a&b are not referred to in the text, we feel that they provide useful information that complements the understanding of Fig. 3c&d.

8. Fig S1, S2, and S4 are never mentioned in the text. Again, are they necessary?

We have removed this section (Change 4).

Reviewers' comments:

Reviewer #1 (Remarks to the Author):

The authors have added an additional section on time integration, which seems to satisfactorily address some concerns of Reviewer 3. I think they have responded satisfactorily to a majority of our other suggestions, though below I detail 4 places where I think they could still do better, thereby improving the paper's readability and reproducibility. All in all, I am still happy to recommend acceptance.

>> p3 bottom right: By "opposite to those in the case $\alpha > 0$ ", do you mean that J^* and t^* are the negative of their values for positive correlations? If so, why would the optimal t^* flip sign when the correlations are negative? [. . .] But I don't understand why changing the sign of the correlation, or the sign of the sensor coupling, would lead you to want to change the sign of t .

> This behaviour results from the fact that the mutual information is invariant under $(\alpha, J, t) \rightarrow (-\alpha, -J, -t)$ which in turn follows from the fact that $(\alpha, J, t, s_2, h_2) \rightarrow (-\alpha, -J, -t, -s_2, -h_2)$ leaves the distributions $P_{\{S|H\}}$ and P_H unchanged (see, Eqs. 4, 7, 8 & 10). Note that S and H are integrated out when we compute the mutual information. We have included this explanation in the revised manuscript (Change 12).

I understand why flipping the sign of coupling is adaptive when the correlation of signals flips. But what is the intuition for the difference made by changing the sign of t ?

>> Give more argument for how you get Eq. 7 from Eq. 2.

> We have included a subsection in the Methods section to discuss the general structure of steady state master equations and the methods for obtaining the solutions (Change 3).

Methods section on Master equation doesn't actually clarify how one gets Eq 7 from Eq 2.

>> Describe how you found the optimal J^* and t^* .

> We now provide an example code for computing the optimal sensor parameters J^* and t^* in the Code Availability section (Change 8).

Making code available to numerically optimize mutual information is nice. But it is also important to describe in the paper (at a high level, perhaps) how you perform this optimization.

Responses to Reviewer 3:

New Discussion sentence "This result highlights sensor cooperativity as a biologically plausible sensing strategy" doesn't seem to actually address reviewer's question.

OPPORTUNITIES FOR FURTHER CLARITY IN NEW TEXT

Fig. 5e: why does the dominant readout number flip to high for negative h , and low for positive h ?

New paragraphs in Discussion: clarify that the cited papers (not you) make the assumption of

continuous deterministic time integration. As written this is not clear.

POSSIBLE TYPOS IN NEW TEXT

Eq 17, rightside equation: Missing minus sign?

Fig 5 caption: readout population should be r , not n

Reviewer #3 (Remarks to the Author):

The authors have significantly improved their manuscript by (i) adding a nontrivial extension to their model consisting of readout molecules that allow them to investigate time integration, and (ii) discussing the relationship between their results and previous results in the field including those on time integration.

As such, I feel that the manuscript has risen to the level necessary for Nature Communications. However, there are a few remaining points that need clarification before I would be comfortable recommending publication:

- With the readout molecules, in addition to more equitably distributing states, introducing nonequilibrium driving seems to have switched the direction of the input-output relation (i.e. Fig 5e goes down-and-to-the-right instead of up-and-to-the-right). This did not happen without the readout (Fig 4d) and is therefore unexpected, yet the authors do not mention it. Why does this happen?

- The authors responded to my question about why they use $I(h_1, h_2)$ instead of α in Fig 2b by saying that it is more natural. However, to me it seems more natural to use the actual parameter instead of introducing a new measure. Part of the reason for the question was to simply understand whether I am missing anything here. Is $I(h_1, h_2)$ monotonic with α ? If so, is the phase diagram of Fig 2b qualitatively the same if one uses α ? I am just not sure what the issue is, if any. Clarification is missing here.

- I pointed out the Fig 3a and b are not mentioned in the text. They should either be (i) removed or (ii) kept and mentioned in the text. The authors preferred to keep them. Therefore, they should mention them in the text. Without a mention, Fig 3a and b are just sitting there awkwardly with the reader wondering why they were never mentioned.

Response to Reviewers 1 and 3

We thank the Reviewers for a careful reading of our revised manuscript and for suggestions which have improved our work. Below we address the remaining comments point by point.

Reviewer 1 — Fig. 5e: why does the dominant readout number flip to high for negative h , and low for positive h ?

Reviewer 3 — With the readout molecules, in addition to more equitably distributing states, introducing nonequilibrium driving seems to have switched the direction of the input-output relation (i.e. Fig 5e goes down-and-to-the-right instead of up-and-to-the-right). This did not happen without the readout (Fig 4d) and is therefore unexpected, yet the authors do not mention it. Why does this happen?

The staircase structure in Fig. 5e corresponds to the distribution of the readout states, associated with the anticorrelated sensor states $+-$ and $-+$, which can correlate with either $h > 0$ or $h < 0$ and either $r > r_0/2$ or $r < r_0/2$, depending on the signs of t and δ . Hence we do not *a priori* expect that r would increase (or decrease) with h .

To clarify this point, we added Supp. Fig. 3 which depicts the conditional distributions $P(S|h)$ and $P(r|h)$ and how t and δ affect the signal-sensor and signal-readout relations. We refer to this new figure in the main text at the end of the subsection *Time Integration*.

Strong positive (negative) signals, i.e., when $|h| > |J_{ij}|$, *always* favour the state $++$ ($--$) and high (low) readouts — see, Figs. 4 & 5c-d. This feature is not visible in Fig. 5e because it shows $P(r, h)$ at weak signals, i.e., $-2.2 < h < 2.2$ while $|J_{21}| = 5.5$. In Supp. Fig. 3, the conditional distribution $P(r|h)$ clearly illustrates the expected signal-readout relation at strong signals for near optimal nonequilibrium sensing in the presence of readout population.

Response to Reviewer 1

I understand why flipping the sign of coupling is adaptive when the correlation of signals flips. But what is the intuition for the difference made by changing the sign of t ?

We have replaced the opaque mathematical relations with a more intuitive explanation which also clarifies that a sign inversion of t relates two equivalent, time-reversed sensing strategies: ‘*The sensing strategies $t = \pm|t^*$ are time-reversed partners of one another both of which yield the same mutual information. This symmetry results from the fact that the signal prior P_H (Eq. (10)) is invariant under $h_1 \leftrightarrow h_2$, hence the freedom in the choice of dominant sensor (i.e., we can either make $J_{12} > J_{21}$ or $J_{12} < J_{21}$).*’

Methods section on Master equation doesn't actually clarify how one gets Eq 7 from Eq 2

We have rewritten a sentence in *Methods* as follows, ‘*For two coupled sensors considered*

in Nonequilibrium coupled sensors, Eq. (16) (Eq. (2)) becomes a set of four simultaneous equations which we solve analytically for a solution (Eq. (7)) that also satisfies the constraints of a probability distribution ($\sum_j p_j = 1$, $p_i \in \mathbb{R}$ and $p_i \geq 0 \forall i$).

Making code available to numerically optimize mutual information is nice. But it is also important to describe in the paper (at a high level, perhaps) how you perform this optimization.

We added the following sentence below Eq. (9), ‘In practice we solve this optimisation problem by a numerical search in the J - t parameter space using standard numerical-analysis software (see, Code Availability for an example code for numerical optimisation of mutual information).’

New Discussion sentence “This result highlights sensor cooperativity as a biologically plausible sensing strategy” doesn’t seem to actually address reviewer [3]’s question.

This sentence includes a reference to Refs. [11-13] and thus places out result on cooperativity in the context of the existing literature.

New paragraphs in Discussion: clarify that the cited papers (not you) make the assumption of continuous deterministic time integration. As written this is not clear.

We have clarified that we make no such assumption in our model.

- Eq 17, rightside equation: Missing minus sign?
- Fig 5 caption: readout population should be r , not n

We thank the reviewer for pointing out these typos. We have corrected them.

Response to Reviewer 3

The authors responded to my question about why they use $I(h_1, h_2)$ instead of α in Fig 2b by saying that it is more natural. However, to me it seems more natural to use the actual parameter instead of introducing a new measure. Part of the reason for the question was to simply understand whether I am missing anything here. Is $I(h_1, h_2)$ monotonic with α ? If so, is the phase diagram of Fig 2b qualitatively the same if one uses alpha? I am just not sure what the issue is, if any. Clarification is missing here.

The relationship between the signal redundancy and correlation is monotonic $I(h_1; h_2) = -\frac{1}{2} \log_2(1 - \alpha^2)$, and thus Fig. 2b remains qualitative the same if we use α instead of $I(h_1; h_2)$. The figure below clearly illustrates the monotonicity between $I(h_1; h_2)$ and α .

I pointed out the Fig 3a and b are not mentioned in the text. They should either be (i) removed or (ii) kept and mentioned in the text. The authors preferred to keep them. Therefore, they should mention them in the text. Without a mention, Fig 3a and b are just sitting there awkwardly with the reader wondering why they were never mentioned.

We have removed Fig. 3**a-b** and updated the references to the remaining panels.

REVIEWERS' COMMENTS:

Reviewer #1 (Remarks to the Author):

The authors have made reasonable responses to both reviewers' remaining questions. I continue to feel quite firmly that the paper merits publication in Nature Communications, and see no need for a further round of revision.

I will note that I personally did not find their response to the joint question ("Fig. 5e: why does ...") to be especially illuminating, perhaps related to the different parameter values (t , δ) and different display (conditional P vs. joint P) in Fig S3 compared to Fig. 5e.

Two more possible typos:

Fig. 5 caption: "as a results"

Fig S3 caption: "sensor states -+ and -+"

Reviewer #3 (Remarks to the Author):

The authors have addressed all of my concerns. I recommend the manuscript for publication.

Response to Reviewer 1

The authors have made reasonable responses to both reviewers' remaining questions. I continue to feel quite firmly that the paper merits publication in Nature Communications, and see no need for a further round of revision.

We appreciate the Reviewer's thorough reading of our revised manuscript and we are pleased that they feel that our paper *merits publication in Nature Communications*.

I will note that I personally did not find their response to the joint question (“Fig. 5e: why does . . .”) to be especially illuminating, perhaps related to the different parameter values (t, δ) and different display (conditional P vs joint P) in Fig S3 compared to Fig 5e.

We now provide a side-by-side comparison of the joint and conditional probability distributions for the same model parameters as in Fig. 5e in a new supplementary figure (Supp. Fig. 3).

Two more possible typos:

Fig 5 caption: “as a results”

Fig S3 caption: “sensor states -+ and -+”

We have corrected these typos.